# Impact of a Local Government Funded Free Cat Sterilization Program for Owned and Semi-Owned Cats

**DOI:** 10.3390/ani14111615

**Published:** 2024-05-29

**Authors:** Jennifer L. Cotterell, Jacquie Rand, Tamsin S. Barnes, Rebekah Scotney

**Affiliations:** 1Australian Pet Welfare Foundation, Kenmore, QLD 4064, Australia; jacquie@petwelfare.org.au; 2The University of Queensland, School of Veterinary Science, Gatton, QLD 4343, Australia; j.rand@uq.edu.au (J.R.);

**Keywords:** sterilization program, animal management Australia, cats, One Welfare, local government, animal welfare

## Abstract

**Simple Summary:**

Free-roaming cats in urban areas are a source of nuisance complaints. In Australia, legislation relating to requirements that cat owners microchip and contain their cats on their property has been largely ineffective in reducing the number of free-roaming cats, because most are strays with no owner. Cats causing nuisance complaints are typically trapped and impounded, but only 7% of cats entering local government facilities are reclaimed by owners, with the remaining either rehomed or euthanized. Many healthy cats are euthanized, negatively impacting the staff involved. In 2013, the city of Banyule in Victoria funded and implemented a free program for cat sterilization, microchipping, and registration. The program was largely targeted at low-socioeconomic suburbs with the highest cat-related complaints and microtargeted at “hot-spots”. Free transport of cats was offered to community members if needed. Stray cats fed by community members enrolled in the program became owned. Over 8 years, 33.0 cats/1000 residents were sterilized in the three target suburbs (average 4.1 cats/1000 per year). Key findings were city-wide decreases in impoundments by 66%, euthanasia by 82%, and cat-related calls by 36% over 8 years, with savings to council of AU $440,660 for an outlay of AU $77,490.

**Abstract:**

In most states of Australia, local governments (councils) are responsible for the enforcement of legislation relating to domestic cats. Traditional methods used for cat management based on trap–adopt or euthanize programs have been ineffective, with cat-related calls and cat impoundments continuing to increase, resulting in many healthy cats being euthanized. This has detrimental effects on the mental health of animal management officers, staff in shelters and council facilities, and cat caregivers. The city of Banyule, Victoria, implemented a free cat sterilization, microchipping, and registration (licensing) program in 2013/14. Initially, it was targeted at three low-socioeconomic suburbs with the highest cat-related calls and intake, and was microtargeted at call locations. An average of 4.1 cats/1000 residents per year were sterilized over eight years. The program included stray cats being fed by caregivers, provided they took ownership. The program was later expanded city-wide. Over eight years, city-wide cat intake decreased by 66%, euthanasia by 82%, and cats reclaimed by owners increased from 6% of intake (2012/13) to 16% (2020/21). Cat-related calls decreased in the target area by 51%, and city-wide by 36%. The council realized cost savings of AU $440,660 associated with reduced costs for cat-related calls to council (AU $137,170) and charges from the contracted welfare agency (AU $303,490), for an outlay of AU $77,490 for sterilization costs. Instead of the traditional management of urban cats, proactive management based on targeted sterilization should be utilized by government and animal welfare agencies in Australia and internationally. These types of programs are effective at reducing cat intake and euthanasia and are cost-effective.

## 1. Introduction

In Australia, outcomes for cats are considerably worse than for dogs in shelters operated by animal welfare agencies and in animal management facilities operated by local governments (council pounds) [1,2]. Return-to-owner rates are usually much lower than for dogs because few impounded cats are identified with a microchip or tag [1]. Because of overcrowding in pounds and shelters, timid or fearful cats and young kittens are often euthanized upon admission [3]. Across Australia, approximately 33% of cats entering shelters and pounds are euthanized, with the worst-performing quartile of local governments euthanizing 67% to 100% of cats [3]. Euthanasia of animals, particularly healthy and treatable animals, has adverse effects on the psychological health of the staff involved [4,5,6,7].

Many stray cats in urban areas are semi-owned and are provided regular food by people who do not perceive the cat as their property [8]. In Australia, 3–9% of adults feed daily one or more cats. These people do not perceive that they own these cats, and they feed an average of 1.5 cats [9,10]. Most are not sterilized and are a substantial source of unwanted kittens. Many of these cats are poorly socialized and are timid or fearful and are therefore at higher risk of euthanasia in a shelter or pound [11].

### 1.1. Domestic Cat Management in Australia

In Australia, municipal councils (local government organizations) are responsible for the management of domestic cats and enforcement of state government legislation in all states and territories, except for South Australia, Tasmania, and the Northern Territory. However, not all councils are equipped to operate animal management facilities (pounds), and many contract out (i.e., outsource) this work to animal welfare organizations. Thus, many welfare organizations are a hybrid model and have one or more contracts to accept either all impounded cats or only cats not returned to owners after the mandatory holding period (stray-hold period). These organizations also accept strays (lost, injured, and unowned) and owner-relinquished cats directly from the public [11].

### 1.2. Legislation and Cat Management in the State of Victoria

Under the Victorian Domestic Animal Act 1994 [12], cats are not allowed to trespass on private property or cause a nuisance, and cats over three months of age must be microchipped and annually registered with the council they reside in. For cats residing in the city of Banyule in 2023/24, the registration cost was AU $108.40 for an entire (intact) cat, and it was AU $39.00 if sterilized [13]. Under the Victorian Local Government Act 2020, councils have the power to make local laws (bylaws) to respond to issues and needs in the broader community. In all councils in Victoria, under local bylaws (ordinances), there are conditions limiting the number of domestic animals to be kept on a property without a permit, and typically, an annual permit is required to keep more than two cats. Many councils also have bylaws which require cats to be contained on the owners’ property either at night or at all times (“leash” laws in USA) [14,15].

In Victoria, Animal Management Officers (AMOs) are employed by each council to enforce legislation pertaining to the Domestic Animal Act 1994 [12], as well as specific council bylaws, such as mandatory containment or sterilization. The standard response by AMOs in Victoria following receipt of a call related to a found cat or nuisance complaint comprises a reactive response, typically lending a cat trap to the complainant or attending the property and trapping the cat. Following trapping, the cat is held at the pound (animal management facility for council) for the legally required timeframe (eight days for a healthy unidentified cat in Victoria) or transferred to a service provider, such as an animal welfare organization. If the cat is not reclaimed by an owner within the legal timeframe, it is rehomed or euthanized. This method of cat control (combined with education) has not been effective in addressing cat-related issues, with many local governments recording steady increases in cat-related calls and impoundments over time [3,16]. Frequently, AMOs can document attending the same properties numerous times over several years, impounding unwanted cats and/or kittens, providing further evidence that current strategies are not working.

Cat-related calls and impoundments are highest in lower socioeconomic areas [10,17,18]. In these areas, residents who own a cat or are feeding a stray cat they did not actively seek to own, simply cannot afford the costs of sterilizing, microchipping, and registering these cats, particularly if multiple cats are being cared for. In all states in Australia, trap–neuter–return (TNR) is illegal under various legislation relating to abandonment, biosecurity, and containment to property [15,19]. A property owner does have the legal right to humanely trap or catch any nuisance cats on his/her property. However, in Victoria, cats found to be “at large” on private property must be handed to an authorized officer (AMO) to be impounded at an animal management facility.

AMOs can find the enforcement of domestic cat legislation challenging, mainly due to the lack of resources and funding to set up effective preventative programs. The lack of effective programs to prevent kitten births, as well as the resulting high euthanasia rates, has a negative impact on the job satisfaction and psychological well-being of AMOs [4].

### 1.3. Cat Management in the City of Banyule, Victoria

The city of Banyule is 19 km from the central business district (CBD) of Greater Melbourne, Victoria, and had a population of 128,832 people in 2021 [20]. The city did not have a bylaw requiring cats to be contained at all times on the owners’ property, but cat owners were subject to state government legislation requiring that their cat not trespass on private property or cause a nuisance. Of the 20 suburbs in the city, three suburbs with a combined population of 14,003 in 2021 were classified as disadvantaged suburbs with Socio-Economic Indexes for Areas (SEIFAs) [21] below 1000 (830, 907, and 981) in 2016 and have more social (government) housing than other areas of the city [22]. These suburbs also had the highest number of cat-related calls to council per 1000 residents (average of 10.8 calls vs 3.5 calls per year for other suburbs in 2011 and 2012 calendar years). The remaining suburbs had SEIFA indices that ranged from 1011 to 1145, with an average of 1055 (average SEIFA across Australia is 1000) [21]. The avenues for subsidized sterilization for community pets in Banyule prior to 2013 were limited to the Australian Veterinary Association (AVA) Desexing (sterilization) Voucher Scheme [23]. This scheme provided residents with a 25% reduction in advertised prices, which varied between veterinary clinics. Anecdotally, many residents still could not afford to pay for the sterilization of their cat/s.

The standard practice for cat management up to 2018 involved AMOs reactively attending the property and providing trap cages to complainants to trap, or AMOs trapped nuisance cats immediately, with no notification to the community that this was occurring. A free relinquishment service was also provided for domestic animals, which, if necessary, incorporated free pickup of the animal. In the financial year 2012/13, the city of Banyule had two full-time AMOs. Trapped cats were transported to the council-contracted animal management facility, the Cat Protection Society (CPS). The CPS was located within the city of Banyule, and residents were also able to deliver stray and relinquished cats or kittens directly to the facility. This was at no cost to the residents, but AU $80 per cat or kitten was charged to council, and this increased to AU $150 in 2018. The CPS had numerous councils contracted to its facility, none of which had effective programs in place to manage their stray cat populations. This frequently saw the facility at or over capacity. Intake was always extreme throughout kitten season, resulting in most kittens under eight weeks of age being euthanized. Due to the number of councils contracting to the CPS, the facility was at capacity all year, with 90% of the 12,000 cats or kittens admitted in 2011 euthanized [24].

### 1.4. Catalyst for Change

A catalyst for change to cat management methods in Banyule occurred in 2012. Specifically, considerable negative psychological impacts were experienced by two AMOs who, in 2012, were forced to deliver a stray socialized kitten to the contracted animal management facility (CPS) only to have the kitten euthanized upon admission because it was under eight weeks of age. That day in 2012, the two AMOs decided that there had to be a better way to manage cats and were no longer willing to continue using the standard trapping practice resulting in the euthanasia of kittens under eight weeks of age, or cats which appeared unsocial at the time of impound, and cats that were not reclaimed or rehomed but were healthy. This method of cat management was clearly not working; many relinquished or stray cats and kittens continued to be collected from the same properties each year, and numerous nuisance cat complaints within the municipality also continued.

A new program for cat management was initiated in response to both the negative psychological impacts on AMOs and shelter staff, and the non-effectiveness of the pre-2013 cat management strategy. The program proposed and approved by the city of Banyule was that sterilization, microchipping, and the first year of registration would be funded by the council. The purpose of this program was to increase ownership responsibilities for owned and stray cats being fed by residents (semi-owned cats) and to reduce unwanted kittens being born and, therefore, the number of cats and kittens killed in the council-contracted facility (CPS). This was provided at no cost for all owned cats and semi-owned cats in the target areas. For the cat to be included in the program, semi-owners were required to take ownership of the cat at the time of sterilization, and their contact details were entered into microchip and municipal databases. Free transport was provided if needed for residents’ cats to and from the veterinary clinic. The trial program was implemented in October 2013 and initially targeted three suburbs where the council received the highest number of cat-related calls, and where enforcement action had resulted in little effect. In addition, to encourage adoptions, the council subsidized free registration for the first year of ownership for any resident of Banyule adopting a cat through the CPS.

The aim of this report is to document the methods and results of this free sterilization and identification program for owned and semi-owned cats, predominantly targeted at suburbs with high cat-related calls in the city of Banyule, Victoria, and microtargeted within those suburbs to locations of cat-related calls.

## 2. Materials and Methods

### 2.1. Overview of the Program

Two free sterilization, microchipping, and registration (licensing) programs for owned and semi-owned cats were provided to residents in the city of Banyule (population 124,711 in 2013) (Figure 1). The first program started in 2013 and was targeted at the three disadvantaged suburbs (combined population 13,445). It was medium-intensity, with 3.0 cats/1000 residents sterilized in the first year in these three suburbs. The program continued until the time of reporting (June 2021), except for two financial years (1 July to 30 June for 2015/16 and 2016/17) when it was suspended, and the second city-wide program was initiated (Table A1). The second program began in 2015/16; it was city-wide, low-intensity (sterilized 1.1 cats/1000 residents in first-year city-wide), and initially operated twice per year. It was the sole program conducted in that financial year and the following year (2016/17). From 2018/19 to 2020/21, both programs operated concurrently and continuously throughout the year, with demand decreasing over this period. Both programs incorporated free drop-off and pickup services between one of the sterilization facilities for any resident across the city who did not have the means to transport their cats.

In both the three target suburbs and city-wide, microtargeting (also known as the “red flag” model) was used, but the sterilization rate per 1000 residents was higher in the target suburbs because of the higher cat-related call rate and greater AMO focus [25]. If AMOs received a call related to a stray or nuisance cat, they assumed that there would be more than one cat at that location and focused efforts around that specific area to enroll cats for sterilization. When the medium-intensity program was recommenced in 2017/18, more intensive microtargeting was conducted by one AMO (JC) and involved revisiting properties with remaining unsterilized cats that were not enrolled in any previous years.

The low-intensity program began in 2015/16, continuing to 2020/21, and was available to all residents in the city of Banyule with cats. Initially, it was offered twice a year in March/April and again in August, prior to the start of kitten season in spring (around September in Victoria). The aim was to facilitate sterilization of cats before the breeding season to prevent kittens being born. The timing aimed to target kittens born or obtained during the preceding kitten season (September to March). Additionally, this city-wide program involved some microtargeting to locations where cat-related calls emanated from.

### 2.2. Complaint Handling

The single-pronged approach for cat trapping changed in 2017–18, whereby, if a complaint was received by council, the trapping process was not commenced until eight days later. The immediate response changed to that of a proactive approach, and an AMO would contact the complainant, gathering further information regarding the issue, including the location of the offending cat. If an address was known, the AMO would attend the property and speak with the cat owner/carer, assessing the cat situation and enrolling any cat/s into the program if they were entire and/or not microchipped and/or not registered. Owners and semi-owners were encouraged to enroll their cats into the program before action to trap any offending cat/s was undertaken. Educating owners and semi-owners, and enrolling their cats in the sterilization program was the highest priority, rather than trapping and impounding the offending cat.

If the complainant could not identify the offending cat owner, a letterbox drop of a “notification of cat trapping” was conducted around the vicinity of the complainant’s address. This letter provided notice of seven days to rectify any issues with cats and included information relating to the legislative requirements pertaining to owned cats, especially the requirement that cats were not allowed to trespass on private property or cause a nuisance. Residents were also provided information on the free sterilization, microchipping, and registration program provided by council. The complainant was notified of this impending trapping action. If the problem persisted after seven days, the AMO would return with a trap, assess the best trapping location, provide guidelines and instruction to the complainant on setting the trap at night (trapping only took place at night), and processes to check the trap and contact council first thing in the morning if a cat had been trapped, so that collection was expedient. In cases where there were several cats, resulting in numerous complaints from one area, in these cases, the AMO would attend the property in the morning and check the traps, transporting multiple cats if required. The trapping notification letter stated if any trapped cat was wearing visible identification, it would be returned to the owner, bypassing the pound.

### 2.3. Advertising the Free Sterilizing, Microchipping, and Registration Program for Cats

In the three suburbs, where the medium-intensity targeted program operated, it was promoted and advertised to residents primarily via the AMOs. Advertising material consisted of flyers, material for door knocks, council newsletters, social media, and media articles and was prepared by an AMO (author JC) in conjunction with council’s communication officer. Areas of concentration for advertising included large government housing estates, local community centers, shopping strips, and community noticeboards. Microtargeting was also undertaken by the AMOs using door knocking and flyer drops to locations where stray and relinquished cats and kittens repeatedly originated, and to locations from which cat-related calls to council emanated. Prior local knowledge of the AMOs and their relationships with the community played a large part in public engagement with the program, particularly communities in the microtargeted areas. The AMOs responded to residents’ requests for assistance for cat-related issues by attending addresses and enrolling owned and semi-owned cats into the free sterilization program.

Advertisements for the medium-intensity microtargeted program encouraged residents who were feeding stray cats or kittens, or who had acquired a new cat or kitten that was not sterilized, to call the council office to enroll their cat/s in the program. In addition, to maximize recruitment, the AMOs capitalized on existing relationships with employees of external organizations who were active in the community and were contacted to assist by referring residents to the program. These organizations’ employees included social workers from the local community centers and other organizations, the Victorian police, and inspectors from the Royal Society for Prevention of Cruelty to Animals (RSPCA).

In contrast, the low-intensity city-wide program was advertised to all residents via the council newsletter that was distributed to every household twice a year and the council’s Facebook page. As the program only operated twice yearly between 2015/16 and 2016/17 (Table A1), a waitlist was developed for the months in between programs to ensure that cats/kittens requiring sterilization were identified and not missed. In the city-wide program, at locations where complaints related to nuisance cats were received, AMOs engaged in some microtargeted advertising of the free sterilization program by door knocking and flyer drops.

### 2.4. Bookings for Sterilization Surgeries

Residents responding by phone to the public flyer or council newsletter would reach the council’s customer service center and provide their contact details. This information would be emailed to the AMO (author JC), who would call the resident to obtain additional information about the cat/cats to be enrolled. This not only provided an opportunity to enroll cats in the program, but importantly, it gave an opportunity to build a rapport and ask questions relating to the cat/cats, for example, if the cats were owned or semi-owned, how many they have, and any other pertinent information. Direct enrolments were also made in the field by the AMOs when interacting with residents as part of their duties.

Bookings were allocated to participating veterinary clinics according to the location of the enrolled resident, and the cat and carer/owner information was emailed to the veterinary clinic by the AMOs. Each clinic would call the resident directly to schedule the cat/s for surgery. This procedure and booking system remained in place throughout the entire program until final data collection in 2021.

No limitations were placed on how many cats could be enrolled in the program per resident or household, nor any specific limiting criteria set for residents to enroll cat/cats. Whilst the program was advertised city-wide as being available during a two-week period twice yearly, in practicality, it took longer for the veterinary clinics to complete all scheduled sterilizations. On occasion, when a resident failed to present their cat/s for scheduled sterilization, a follow-up property visit by the AMO was undertaken, along with the veterinary clinic attempting to contact the resident by phone to reschedule the sterilization.

### 2.5. Sterilization Surgeries

From October 2013 to August 2014, surgeries (ovariohysterectomy or orchidectomy (testicular removal) were undertaken by an RSPCA mobile veterinary clinic located centrally within the targeted suburbs. It was staffed with a team of veterinary nurses and two veterinarians. Free cat transportation was provided by AMOs for any resident who had limited or no transport to minimize the barriers for cats to enrolled.

Learnings from the 2013 program resulted in a change to scheduling of services in 2014. Firstly, female cats required longer anesthetic recovery time than the males, and as all cats were being discharged the same day to recover at home, the females were scheduled for sterilization in the morning and males in the afternoon. Females who were heavily pregnant upon arrival at the mobile clinic were not sterilized at that time, but they were sterilized along with the kittens, when eligible.

In 2013, non-dissolvable external skin sutures were used for female cat sterilizations, and AMOs would attend home properties to remove the sutures 10 days after surgery. This proved to be a challenge with shy or fearful owned and semi-owned cats, as even with good cat-handling skills, it was difficult to achieve effective restraint for suture removal. In 2014, dissolvable external skin sutures were utilized, and Elizabethan collars were sent home with cats to minimize potential suture/wound issues.

Complications associated with the surgeries arising after-hours, typically issues with sutures, were referred to local veterinary clinics for management. As a result, it was agreed to be more appropriate to utilize local veterinarians to undertake the sterilizations, as they could then respond to any issues that arose after-hours. 

Two local veterinary clinics located in the south of the municipality in the vicinity of the three target suburbs agreed to participate in the sterilization program from 2015. Both clinics had existing relationships with the council for the emergency treatment of stray animals. Thus, they had an existing rapport with AMOs and cat rescue groups within the area and were eager to support the program.

Between 2018 and 2021, sterilization surgeries were undertaken by a single original veterinary clinic in the south of the municipality, and the veterinary clinic associated with CPS located in the north of the municipality. Having northern and southern clinics made transport easier for residents across the municipality, but transportation remained available to any resident who had limited or no transportation.

Cat traps were sometimes needed to capture poorly socialized semi-owned cats which were unable to be handled. This required flexibility of both the veterinary clinic when scheduling the sterilization procedure and the AMOs to provide transportation. The clinics on occasion needed to reschedule surgeries to fit in with the trapping process, to prioritize the cat’s welfare, and limit time in a trap to reduce stress on the cat.

In late 2018, the council service agreement with the animal management facility (CPS) was renewed for a three-year period. The new agreement included a price increase for a cat or kitten from AU $80 to AU $150. The increase in price negatively impacted the council because it applied to all stray cats and kittens impounded directly by an AMO or directly transported by a resident to CPS, as well as any cat or kitten relinquished by a resident. To compensate for the increased service costs that occurred without warning or future budget planning by council, 200 free sterilization surgeries at CPS were negotiated by one AMO (JC) for residents’ cats enrolled in the free cat sterilization program.

### 2.6. Resources

From 2013 to 2017, two full-time AMOs were available to liaise with residents and staff to schedule surgeries and transport cats to and from the mobile RSPCA Mobile Veterinary Clinic (2013–14) or the two veterinary clinics (2015–2018), using council vehicles equipped for animal transport. The AMOs also conducted door knocking and letterbox flyer drops and liaised with all external organizations. The AMOs operated this program whilst maintaining all other usual day-to-day cat- and dog-related calls and impounds. From 2018 to 2021, one full-time AMO (JC) was available for liaising with residents and veterinary clinics, scheduling surgeries, and, if needed, there was a second AMO to assist with transporting cat/s for surgery. From 2015 to 2021, most residents were able and willing to transport the cat/s to the veterinary clinics.

### 2.7. Costs to the City of Banyule for Cat Management

The calculated cost of a cat-related complaint was based on estimated labor costs to Animal Management Services of AU $204 per cat-related call that was acted on by attending the property, trapping the offending cat, and delivering the cat to CPS (Table A2). This was based on 1 h per call for a customer service officer to take the call and create a job (Band 4A wage @ AU $32.80/h); 1 h for local laws administration to process the call (@ AU $38.89/h); and 3.67 h of AMO time to attend the property, on average, three times (Band 5A wage @ AU $35.89/h; Banyule Enterprise Bargaining Agreement 2017–2021). In addition, vehicle costs of AU $86 were calculated at 40 km per round trip for three trips totaling 120 km at AU $0.72/km [26]. The total labor and vehicle costs of AU $290 per cat-related call were taken into consideration when the call was first logged at council with a customer service officer and then transferred to the local laws team, and it included the time from initial AMO contact to all actions associated with the complaint, including but not limited to the initial phone contact with the resident, travel time and fuel, site visits, and trapping and impounding a cat. In addition, there were costs charged to the city of Banyule under their agreement with CPS and were associated with care in the animal management facility for the eight-day mandated hold period (initially AU $80 per cat increasing to AU $150 in 2018). The same costs applied to the council for any cat/ kitten directly relinquished by the owner or a stray brought by a resident.

### 2.8. Data Management and Analysis

Data for cat-related calls were routinely recorded by the council and obtained directly from council records within the customer request management system between 2011 and 2021. Impound data for cats from 2011/12 to 2020/21 were obtained from the council impound register, maintained by administration officers directly within the animal management department. Data for owner-relinquished cats and stray cats brought directly to the shelter were routinely collected by CPS. However, they were only available from CPS from 2017/21 and were based on electronic records, because paper records prior to 2017 were destroyed at the time of transferring to electronic records.

For the calculation of cost savings to the city of Banyule resulting from fewer cats being admitted to CPS over the eight years of the program, the cost per cat or kitten charged by CPS was AU $80, which increased to AU $150 in 2018/2019. Prior to 2017, only numbers of impounded cats were known, but the total number of cats admitted to CPS from Banyule for those years were estimated using the average proportion of all cats admitted to CPS from Banyule that were impounded for the four years that this was known. Impoundments were 34% of the total cat intake over 2017/18 to 2020/21 (range 29–41%), and based on this, total cost savings over the eight years was calculated.

Data for numbers of sterilization surgeries performed each year were obtained directly from council records. For values shown per 1000 residents, Australian Census data from 2011, 2016, and 2021 were used for the population year in which they were collected, and the following three years until the next census data were available (Table A1) [20].

Ancillary statistical analyses were conducted for two time periods. Firstly, to explore changes over the eight years of the program, 2012/13 values were used as a baseline for analyses by financial year through to 2020/21 for the number of cats impounded, reclaimed, rehomed, and euthanized, and for the percentage of impounded cats that were reclaimed, rehomed, and euthanized. The average of 2011 and 2012 values for cat-related calls and costs were used as a baseline for analyses by calendar year through to 2021 [20]. Secondly, to explore changes over time after the city-wide and targeted-area programs were operated concurrently and trapping policy was modified, 2016/17 was used as a baseline for analyses by financial year through to 2020/21 and 2016 as a baseline for analyses by calendar year through to 2021. Kendall’s tau was used to measure the strength of the relationship between year and the same set of variables. Where data were available at both city-wide and targeted-area levels, analyses were conducted separately for each level. Statistical analyses were performed using Stata 18.0 [27]. The results are shown in Table A3.

## 3. Results

### 3.1. Cats Sterilized per 1000 Residents and Impounded, Relinquished, and Found Stray Cats

In the first financial year of the medium-intensity program (2013/14), 3.0 cats/1000 residents were sterilized across the three target suburbs, equating to 0.3 cats/1000 residents across the city of 125,107 residents (Table A1). In the target suburbs (13,445 residents), the sterilization rate peaked at 5.5 cats/1000 residents in 2014/15 and averaged 4.1 cats/1000 residents over the eight years of the program. When the low-intensity targeted city-wide program was introduced in 2014/15, 0.7 cats/1000 residents were sterilized across the city (eight-year average, 0.8 cats/1000 residents city-wide). Over the eight years of the program, a cumulative total of 33 cats/1000 residents were sterilized in the three target suburbs compared to 6.5 cats/1000 residents city-wide, which included those cats sterilized in the three target suburbs (Table A1).

City-wide, over the eight years of the program to July 2021, the number of cats impounded by AMOs decreased by 66% from 396 (3.2 cats/1000 residents) to 134 (1.1 cat/1000 residents) (tau-b −0.72; *p* = 0.009; Table A1 and Table A3). Importantly, city-wide over the eight years, the number of impounded kittens under 12 weeks of age (legal age for registration) decreased by 75% (85 to 21 kittens; tau-b −0.76; *p* = 0.006).

When the medium-intensity targeted program resumed in 2017/18, and the trapping process changed from enforcement-orientated to assistive, cats impounded city-wide decreased by 51% over four years, from 284 in 2016/17 to 134 in 2020/21 (tau-b −1, *p* = 0.028), which represented a decrease in impounded cats from 2.1 to 1.1 cat/1000 residents.

City-wide, the number of cats and kittens relinquished by owners directly to CPS decreased by 50% (204 to 102) over the final four years of the program when data were available (Table 1). The number of stray cats found trespassing on private property and handed directly into CPS by residents also decreased by 28% (315 to 228) city-wide over the same period. During this time from 2017/18 to 2020/21, 5.4, 3.6, 5.1, and 3.2 cats/1000 residents were sterilized in the target suburbs compared to 1.4, 0.7, 0.7, and 0.6 cats /1000 residents city-wide (including target suburbs) (Table A1).

### 3.2. Outcomes—Return to Owner, Rehomed, and Euthanized

City-wide, the proportion of impounded cats that were reclaimed by owners more than doubled from 6% in 2012/13 to 16% in 2020/21 (Table 2). Although the proportion of impounded cats that were rehomed did not change markedly from 59% at baseline in 2012/13 to 66% in 2020/21 (tau-b 0.00; *p* > 0.999; Table A3), the number rehomed decreased by 63% because substantially fewer cats were impounded, and therefore, fewer cats were available to be rehomed into new homes (tau-b −0.61; *p* = 0.029; Table A3).

The number of impounded cats euthanized decreased by 82% over the eight years of the program, from 138 cats or 1.1 cats/1000 residents at baseline to 25 cats (0.2/1000 residents) in the final year (Table 2; Figure 2). The greatest decrease was realized in the last three years of the program when the cumulative total of cats sterilized in the final year reached 33 cats/1000 residents in the three suburbs and 6.5 cats/1000 residents city-wide. The response to nuisance-cat complaints had also changed to an assistive response (Table A1).

### 3.3. Cat-Related Complaints and Found Cat Calls to Council

Calls to the council relating to found cats and nuisance cats decreased city-wide by 36% from 436 at baseline (3.5 calls/1000 residents averaged over 2011 and 2012) to 281 (2.2 calls/1000 residents) in the final year of the program (Table 3 and Figure 3). Importantly, in the three target suburbs, cat-related calls decreased by 51% from 10.8 calls/1000 residents (average over 2011 and 2012) to 5.1 calls/1000 residents (tau-b −0.67, *p* = 0.009; Table A3).

### 3.4. City-Wide Savings in Costs for Impoundments and for Cat-Related Calls versus Costs Incurred for the Sterilization Program

Estimated cost savings over 8 years of AU $440,660 were realized resulting from decreased cat-related calls to council (AU $137,170) and decreased charges from the contracted welfare agency (AU $303,490), for an outlay of AU $77,490 for sterilization costs (Table 4).

The average cost of sterilizations to the city of Banyule was AU $105 per cat/kitten (80 cats @ AU $157.50 charged by RSPCA, 618 cats @ AU $105 charged by local veterinarians, and between 2018 and 2021, 133 free sterilizations were utilized out of the 200 free sterilizations by CPS that were negotiated with the contract renewal). A total of 831 cats were sterilized over the eight years of the program. The total cost from 2013 to 2021 for sterilizations and microchipping was AU $77,490.

A reduction of costs to the city of Banyule of AU $137,170 over the eight calendar years of the sterilization program emanated from reduced AMO time and vehicle costs for attending to cat-related calls from the community. This was based on AU $203 for each cat-related call to be processed by council staff and AMOs to act on a call (total staff time 5 h 40 min), plus vehicle costs of AU $86 (average of 120 kms travelled for 3 site visits), with a total cost per call of AU $290 (Table A2). Minimal additional costs were incurred for staff time if the cat-related call resulted in the cat being enrolled in the sterilization program rather than being impounded, and these costs were not considered.

There were cost savings of AU $104,120 associated with reduced numbers of cats impounded by AMOs over the eight years of the program. The city of Banyule was charged under their agreement with CPS for care of impounded cats in the CPS shelter for the eight-day mandated hold period (initially AU $80 per cat increasing to AU $150 in 2018). Although the same costs were charged to the council for any cat or kitten directly relinquished by the owner, or a stray brought by a resident, these data were not available prior to 2017. However, these additional savings to council were estimated at AU $229,267 because impounded cats represented an average of 34% of cats emanating from the city of Banyule into CPS during the period when data were available (2017/18 to 2020/21). Therefore, the total cost savings associated with cats entering CPS from Banyule from the start of the program would likely be in the order of AU $303,490.

The Cat Protection Society would also have realized reduced costs because they incurred costs associated with either euthanasia or veterinary care, sterilization, microchipping, and rehoming of the cat after the eight-day hold, and these were not charged to the council.

## 4. Discussion

The city of Banyule (population 124,711 in 2013) implemented a free sterilization, microchipping, and registration program in October 2013 for cats that were owned or stray cats cared for by residents (semi-owners). The aim was to trial a new way to manage cat-related issues instead of continuing solely with a traditional trap–rehome or euthanize policy. Initially, a medium-intensity program (3.0 cats sterilized/1000 residents) was implemented in three suburbs with the highest numbers of impounded cats per 1000 residents and cat-related calls to council, and was microtargeted at the location of calls related to nuisance cats and found cats. Transportation was offered for residents unable to transport the cats they were caring for. In 2014/15, a low-intensity city-wide program (0.7 cat sterilized/1000 residents) was implemented, and the targeted program suspended until 2017/18, when both programs operated concurrently. Over the eight years, a cumulative total of 33 cats/1000 residents were sterilized in the three target suburbs (average 4.1 cats per year), and 6.5 cats/1000 residents city-wide (average 0.8 cats/1000 per year). Whilst the cat trapping continued throughout the 8 years, in 2018 the focus shifted from immediately providing a cat trap and trapping the nuisance cat to attending the property of the cat owner and/or complainant, discussing the complaint lodged, and if possible, enrolling the offending cat and any other entire cats into the sterilization program. Key findings over the 8-year program were that city-wide, impoundments decreased by 51%, euthanasia by 82%, cat-related calls by 36% and calls in the target area by 51%. Cost savings to council were estimated at AU $440,660 for reduced staff time, travel costs and charges from CPS, and council outlaid AU $77,490 for sterilization costs.

### 4.1. Cats Sterilized and Impoundments

Following reinstitution of the medium-intensity program targeted at the three suburbs in 2017/18, and a change in trapping policy, intake into the shelter from the city of Banyule, decreased by 68% from 6.1 (2017/18) to 3.6 (2020/21) cats/1000 residents, compared to the average for Victoria of 7.2 cats/1000 residents in 2018 [3]. City-wide, impoundments decreased by 49% from 2.1 to 1 cats/1000 residents, owner-relinquished cats by 50% and stray cats brought in by members of the public by 28%. After sterilizing an average of 4.3 cats/1000 residents for four years (2017/18 to 2020/21), the combined reduction in intake from all sources was 47%. This is consistent with reports from six cities in USA ranging in population from 200,000 to 1.8 million where an average of 5.4 cats/1000 residents were sterilized each year over three years, and shelter intake decreased by a median of 32% [25]. The greatest decrease in intake occurred in Columbus which implemented the “red flag” model where animal control officers utilized prior knowledge of “hot-spot” areas to target sterilizations, reducing intake by 45% [28]. This was similar to the method used in Banyule where AMOs microtargeted locations known to be a source of impounded cats and cat-related complaints.

A higher-intensity program in Florida that was targeted at an area of nearly 20,000 residents with high shelter intake of cats, sterilized 60 cats/1000 residents per year for two years, but it did not focus on microtargeting. Shelter intake from the target area decreased 30% in the first year, and 66% over the two years [29]. This compares very favorably to the 41% reduction over the last three years of our program for all sources of cats entering the CPS facility from the whole of the city of Banyule, and a 57% reduction in impounded cats from the target area in the last 2 years after sterilizing approximately 4.0 cats/1000 residents each year, using a microtargeted approach. In Florida, a total of 2366 cats were sterilized over the two years, and 52% were returned to the original location or relocated to other sites after sterilization (TNR), whereas 47% were sterilized and rehomed or transferred to a rescue group and were not included in the shelter intake in the target area [29].

Based on data from our study, resolving complaints by providing the owner or semi-owner whose cats were a source of complaints an opportunity to sterilize the cats they were caring for at no cost, rather than immediately deploying traps as traditionally done, is an effective way to reduce the shelter intake of cats and council costs. It is consistent with an assistive approach rather than traditional punishment-orientated animal control and is aligned with the One Welfare philosophy, which aims to balance and optimize the well-being of humans, animals, and the environment [30].

Key differences between the Banyule program and the programs described in the 6 US cities [25] and in Florida [29] were that cats in the USA were sterilized and returned to their outside homes without the carer being required to take ownership of the cats, commonly called trap, neuter, and return (TNR). Although proven successful, TNR is illegal in all states in Australia [31,32]. In the Banyule program, where there were multiple cats on private property and there was an identified caregiver, cats were sterilized and returned, but they were microchipped and registered to the caregiver, who became the legal owner. Thus, semi-owned cats became fully owned, with permanent identification. In the Banyule program, most of the cats sterilized were owned, with fewer semi-owned cats sterilized, including a minority from multi-cat sites (>3–8 cats) on private property. The decrease in cat intake associated with sterilizing owned cats is consistent with recent modeling showing that sterilizing owned cats in the UK will have the greatest impact on decreasing the free-roaming cat population [33].

The initial shelter intake in the Florida study was 14 cats/1000 residents [29]. Although Banyule data were not available prior to 2017/18 for owner-relinquished cats and strays admitted directly by the public to the shelter, based on known impoundments from the target area at baseline and intake of impoundments, strays from the general public and owner-relinquished cats over the last four years of the program, the total shelter intake from the three target suburbs was estimated to be around 23.5 cats/1000 residents at baseline in 2012/13 (city-wide, 9.4 cats/1000 residents) and approximately 15.9 cats/1000 residents in 2016/17.

Our results demonstrate that marked decreases in shelter cat intake can be achieved over four years and eight years of providing a free sterilization, microchipping, and registration program for owned and semi-owned cats fed by residents, and by changing from an enforcement-based response to cat-related complaints to an assistive approach. By microtargeting the program toward the cats that are most at risk of impoundment or surrender, the decrease in intake was comparable to that achieved in Florida in their target area, but achieved at much lower levels of sterilization (4.3 cats/1000 residents/year for four years versus 60 cats/1000 residents/year for two years) and, hence, lower cost. The reduction in cat intake freed up resources of the contracted shelter (CPS) and enabled them to assist other animal welfare shelters by taking their excess cats and kittens for rehoming.

Many Victorian councils already provide low-cost or subsidized sterilization programs for cats. There are concerns that these schemes primarily provide a cheaper option for owners who were already going to sterilize their cats [34]. Sterilization programs need clear goals, targeting the locations that have the highest cat-related complaints and/or shelter and pound intake if they are to have measurable effects in reducing costs and reducing the intake and euthanasia of healthy and treatable cats and kittens [35,36].

### 4.2. Outcomes—Reclaimed by Owner, Rehomed, and Euthanized

#### 4.2.1. Reclaimed

Over the eight years of the program, the proportion of cats reclaimed by owners city-wide more than doubled from 6%, which was just below the reported average (7%) for Victoria in 2018 [1,3], to 20% in 2019/2020 and 16% in 2021/22. Microchipping is mandatory for registration (licensing) in Victoria, and all 831 cats sterilized in the Banyule program were implanted with a microchip for permanent identification. The Domestic Animals Act 1994 requires councils to pay the State Government Treasurer AU $4.16 for every cat registration, and the city of Banyule was still obligated to pay this mandatory State Government levy to the Treasurer, even if they provided free or discounted registration for residents’ cats. However, in other states, such as NSW, this is a bigger barrier to councils implementing free or low-cost sterilization, microchipping, and registration programs because the whole fee for registration and other permits has to be paid to the State Government, which, in NSW (from 1 July 2023), would represent AU $157 per cat if it was older than four months (AU $65 lifetime registration plus AU $92 if cat older than 4 months when sterilized). Legislative changes are required to minimize extra costs to councils and facilitate programs to get cats sterilized and identified so that they can be returned to owners if wandering or escaped.

It was considered best practice that all stray cats collected by AMOs were scanned in the field (when safe or possible to do so) for a microchip to facilitate cats being returned directly to their owner. In cases where the owner could be quickly contacted, it avoided impoundment of a wandering owned cat. Increasing the proportion of cats returned to owners decreases the proportion that require to be rehomed. Hence, returning cats to owners decreases shelter intake and costs and improves cat welfare, including reducing the risk of shelter-acquired infectious disease [37].

In TNR and return-to-field (RTF) programs, cats are not traditionally implanted with a microchip, and these cats are sterilized and returned to where found. Because TNR and RTF are illegal in Australia, where multiple cats were sterilized for a semi-owner (cat caregiver), their name was lodged on the microchip and registration databases. Along with sterilization, TNR programs in the USA focus on vaccination for rabies, rather than identification by microchipping. In contrast to the USA, Australia is rabies-free, and microchipping is mandatory in all states and territories, except the Northern Territory.

#### 4.2.2. Adoption of Stray Cats

The total number of cats rehomed decreased by 66%, although the percentage of impounded cats that were rehomed did not change significantly from baseline (66% versus 59% at baseline). This was because the number of cats impounded was greatly reduced. Decreasing the number of cats requiring adoption represents a significant reduction in costs for the shelter, which are reported to be AU $1540 (AU $385 per week) for a 30-day length of stay, including the veterinary care required before rehoming [11]. It also frees up shelter resources which could be invested in sterilization programs, pet retention programs, or extending services to other areas [38,39,40].

For shelters with contracts with local government animal management services, cats brought in by AMOs that are typically trapped in response to nuisance complaints were reported to place a greater burden on the shelter because of their lower adoption rates and longer length of stay than cats relinquished directly to these facilities by finders, owners, or semi-owners [41]. In Illinois, stray cats impounded by AMOs at a shelter took longer to rehome than a stray cat brought in by a finder (68 versus 61 days, including the seven-day holding period). Owner-relinquished cats spent the least amount of time in the shelter before rehoming, associated with their greater sociability (48 days) [41].

#### 4.2.3. Euthanasia

The number of impounded cats that were euthanized decreased by 82% over the eight years of the program. This reflected a decrease in the total number of cats impounded, as well as a decrease in the percentage euthanized, from 35% (138/396) to 19% (25/134). The number of cats euthanized city-wide decreased from 1.1 cats/ 1000 residents in 2012/13, to 0.2 cats/1000 residents in 2020/21. Subjectively, AMOs observed that the sociability of impounded cats and kittens improved over the eight years of the program. There was a noticeable reduction in the proportion of trapped cats and kittens that were living under houses or born in sheds, and an increase in the proportion raised in home environments and handled.

This decrease in euthanasia is consistent with the results from six US cities where a median reduction of 83% in the numbers of cats euthanized occurred after sterilizing an average of 5.4 cats/1000 residents per year for three years [28]. However, in contrast to the Banyule program, which sterilized mostly owned cats in low-socioeconomic suburbs, the US programs employed TNR [42] and return-to-field (shelter–neuter–return) practices [43].

Return-to-field (RTF) programs, also called shelter–neuter–return (SNR), involve unidentified cats that are trapped by AMOs or brought in as strays from the public and are healthy, but may not be readily rehomed because they are fearful of people [41]. These cats are sterilized, vaccinated, ear tipped, and returned to their outdoor home location. This is based on the premise that someone is feeding them, and that, in the shelter, these cats have lower adoption rates and higher euthanasia than cats from other sources [41]. Following the implementation of RTF programs, cat and kitten euthanasia decreased by nearly 50% (from 70% to 23%), impounds decreased by 29%, respiratory disease in the shelter decreased by 99%, and dead-cat pickup off the streets declined by 20% [37,44]. The return-to-field practice is increasingly being embraced in the USA, and in 2023, 6.6% of cats were returned to field, representing 12.6% of stray-cat intake [2].

By reducing the intake of poorly socialized cats, TNR also reduces the number of cats euthanized for behavioral reasons [29]. TNR was introduced in the USA because the traditional methods of cat management based on trapping, impounding, rehoming, or euthanizing were not effective in stopping complaints or impoundments and were increasingly unacceptable to the community and shelter staff because of the high numbers of healthy cats and kittens being killed [45]. In contrast to Australia, where TNR is illegal in all states and territories, TNR is legal in most states of the USA.

In Australia, the average euthanasia rate for shelters and municipalities operating their own facility (excluding rescue groups) was 33% in 2018/19, and a quarter of municipalities operating their own facilities in Victoria euthanized 73–98% of cats [3]. The impact of high euthanasia rates significantly contributes to the poor mental health of AMOs, greatly reducing job satisfaction and psychological well-being. The potential for psychological injury is not limited to AMOs, but effects all animal carers and advocates who are exposed to euthanizing healthy and treatable cats, including the veterinarians and shelter workers who provide care, form attachments, and are bonded to the cats [1,4,7,46,47].

AMOs in the city of Banyule observed an increase in job satisfaction associated with offering the free cat sterilization, microchipping, and registration program to those in the community who had cats they owned or cats they were caring for (per comm JC). This resulted in a feeling of being empowered to provide proactive solutions to the community for cat-related issues, rather than engaging in reactive measures which resulted in feeling helpless and unable to effectively assist residents or their cats. The program increased rapport with the community and contributed to improved job satisfaction for the AMOs.

The benefit to council in making cat management programs more proactive and assistive, and less enforcement-centered and reactive, is that it lowers the risk of psychological injury to its AMOs, shelter and pound staff, and veterinarians in private practices with council contracts. This may not only increase staff retention and provide greater job satisfaction, but also positively impact council expenditure relating to absenteeism, workers compensation, and recruitment [1,48]. A US study on the impacts of euthanasia on employee turnover in animal shelters found that making decisions regarding euthanasia of animals on the basis of factors other than health and behavior reasons was associated with higher staff turnover [7].

Animal and human welfare are connected, and by improving animal welfare, human welfare is improved, and vice versa [49,50]. Engaging in an assistive-centered approach to cat management aligns with a One Welfare framework. This is characterized by a collaboration between AMOs and the community that focuses on optimizing the well-being of humans, animals, and their environments. Shelters and municipalities that embrace free sterilization programs for owned and semi-owned cats, microtargeted at locations of high cat impoundments or cat-related calls, can contribute to greater animal and human welfare. Aligning cat management with One Welfare is likely to provide more successful long-term outcomes for owned and semi-owned cats and for those who care for them [46,49].

It is recommended that legislation in Australia be amended to allow RTF and TNR to reduce the number of healthy and treatable cats killed in shelters and municipal facilities and negatively impacting staff. When applied with sufficient intensity, TNR will also reduce the environmental footprint of free-roaming cats, because it effectively decreases the number of cats over time [31,32]. Given that animal welfare, human well-being, and environmental conservation are inextricably linked [49], these recommended changes to legislation and cat management practices would contribute to addressing the adverse impacts on the psychological well-being of veterinarians, shelter staff, and AMOs, as well as benefiting the environment. In the Banyule program, stray cats fed by caregivers (semi-owners) could not be sterilized if the caregiver was unwilling or unable to take ownership of the cats. To sterilize these cats, which may involve multiple cats on private property, around businesses, or in public places, it will require legislative changes to be legal. In Banyule, all carers of semi-owned cats accepted ownership of these cats when proposed by the AMOs. However, mandated containment bylaws requiring cats to be contained to the owner’s property can be a barrier to semi-owners taking ownership of stray cats, because rental properties typically have inadequate fencing and may lack screens on windows or doors, and for low-income semi-owners, costs are a barrier for containment systems.

### 4.3. Complaints/Found Cat Calls to Council

Calls related to found cats and nuisance cats decreased city-wide by 36% over the eight years of the program, and by 51% in the target areas, which, at baseline, had substantially higher calls to council relating to cats. This reduction in complaints is consistent with that observed in the USA in regard to sterilization programs [35,51]. However, there was a transient increase in complaints in 2020, potentially attributable to COVID-19, when residents in the city of Banyule were in lockdown and working from home in 2020, from mid-March to the end of October [52]. Because residents were working from home during the pandemic, it is possible they were more aware of free-roaming cats and had more time to make complaints to council during 2020 [53]. Complaints decreased again in 2021.

During the lockdown, except for a seven-day isolation period following a staff member’s positive COVID test, AMOs worked in the field, addressing complaints and, if necessary, trapping cats. During the 7.5-month lockdown, staff undertook paperwork at home because the council office was closed for resident and staff access. Except for the seven-day isolation, AMOs were able to deliver impounded cats to CPS, as well as stray and owner relinquished cats, but the general public was not permitted access. There was also an initial closing of veterinary clinics, which halted the free sterilization program, but this was restarted after a four-week period, when veterinary services were deemed essential services, and reopened with limitations for access by residents [54].

Current cat management strategies for AMOs in Australia and many other countries are focused on enforcement action, rather than having a One Welfare approach [30]. This limits the ability to resolve issues relating to cat management in the long-term, because it does not consider the triad relationship between people, their animals, and the environment. For example, it is assumed easier to serve an enforcement notice on a resident directing him/her to keep a cat confined to a property and have it sterilized and registered (a reactive approach), rather than to provide an avenue for education and proactively assist a resident to achieve suitable and appropriate cat management objectives. The enforcement focus is on non-compliance rather than garnering an understanding of community or individual needs. Cat intake into shelters and pounds is highest from low-socioeconomic areas [55], where residents are less able to comply with enforcement orders because of barriers of cost and accessibility. Proactively working with the community to overcome barriers to providing good animal health and welfare will ensure not only better outcomes for people and animals, but it will achieve greater levels of compliance [56].

Implementing a targeted sterilization program proved to be a more effective strategy for the city of Banyule than introducing further by-laws, for example, mandated containment or sterilization of cats, particularly in low-socioeconomic areas, where compliance would be minimal for many cat owners or semi-owners because of the cost barrier [56]. With a substantial proportion of free-roaming cats in urban areas being semi-owned, it is impossible for further mandates to be complied with in the community where there is no owner or an owner cannot be identified. Many councils in Victoria have mandated containment, but because most trapped cats have no identification and, hence, no identifiable owner, mandatory confinement is, in most cases, not enforceable [57,58,59] and results in many healthy cats being trapped and euthanized. Before further cat mandates are implemented, councils should consider how the mandate aligns with council objectives, consider how success will be measured if implemented, and, most importantly, consider alternatives to imposing a more restrictive local law on residents who may not be able to comply. These are part of the guidelines that councils are encouraged to consider when preparing to make a local law, along with ensuring the least burden with the greatest advantage to community members [60]. It is important that there is an effective consultation process with stakeholders, and evidence-based information is provided on the reasons for free-roaming cats, effective management methods to reduce their impact, and the impact of proposed bylaws on disadvantaged residents. In the USA, in many areas where cat confinement laws were introduced, they have since been retracted, because compliance is impossible to enforce or achieve when there is no identified owner [61,62].

In Victorian councils and many others across Australia, traditional trap, impound, reclaim, adopt, or euthanize methods are currently utilized. However, there are many stakeholders with differing opinions relating to cat management strategies. Community stakeholders and the general public have an expectation that animal welfare legislation meets their community expectations, and in the past, these expectations have been a major driver for legislative changes relating to animal welfare [16,63]. This is aligned with the concept of Social License to Operate (SLO), which refers to the implicit process by which a community gives approval to conduct activities [64].

The traditional method for cat management is becoming increasingly unacceptable in the community and with staff because of the continuing euthanasia of healthy cats and kittens. Councils would be wise to evolve by shifting strategies to more proactive engagement with the community and stakeholders, ensuring that the interrelationships between animal welfare, human well-being, and their physical and social environments are strengthened.

Employing the One Welfare framework to cat management strategies assumes an assistive-centered approach rather than an enforcement-centered approach. The One Welfare concept recognizes that, while animal welfare and human well-being are interlinked, there is a need for both to be considered in a social context, considering the mental health benefits of owning an animal and negative impacts on staff, including veterinarians, and on cat carers of euthanizing healthy animals [4,50].

### 4.4. Costs Associated with City-Wide Impoundments and Cat Nuisance Calls versus Costs for Sterilization

Council savings of AU $137,170 over eight years were realized associated with reduced cat-related calls and the associated reduction in vehicle costs and time spent by AMOs addressing complaints (estimated at AU $290 per call). However, the greatest savings were related to reduced costs charged by CPS to council over eight years for cats emanating from the city (AU $80/cat increasing to AU $150/cat), estimated at AU $303,490. The total estimated savings over eight years were AU $440,660. In comparison, the total cost to sterilize 831 cats was AU $77,490.

A study from Salt Lake City, US, in 2020, reported an average of US $400 per animal to implement an enforcement approach, which included officer response to attend, veterinary care, shelter housing, and rehoming costs [56]. This was similar to the costs for the city of Banyule from 2019 of AU $440 (AU $290 plus AU $150). Based on these data demonstrating cost savings and the benefits to staff of reduced euthanasia and to the community from fewer cat-related complaints, it is recommended that councils and animal welfare organizations invest in a more sustainable model of support, aimed at keeping animals and their caregivers together [56].

The cost to the city of Banyule per impounded cat of AU $80 to AU $150 was at the lower end of pricing from service providers in Victoria, with some councils paying up to AU $500 per impounded cat for housing for the mandated eight-day hold. However, many council service agreements only require payment for impounded cats brought in by AMOs, and not for owner-relinquished cats or stray cats coming directly to the shelter from the public. This was in contrast to the service agreement between CPS and the city of Banyule, which required all cats emanating from the local government area to be paid for. However, the city would have paid a similar total amount if charged AU $450 only for impounded cats.

In a study from USA in 2022, a computer simulation model was utilized to estimate and compare costs for management options for free roaming cats to reduce the population size by 45% over 10 years [65]. Costs were calculated while assuming that 75% of cats would be trapped and managed in one of five ways—trapped and euthanized; trapped, sterilized, and all cats returned to location (TNR); TNR with 10% of trapped cats rehomed (mainly kittens); trapped and, after a mandatory hold of 7.5 days in shelter, all rehomed; or half rehomed and half euthanized. The greatest cost was seen with trapping and adopting all cats (average cost US $342 per cat). Trap, hold in shelter for eight days, and if not reclaimed by an owner, adopt or euthanize, is the method commonly used across councils in Victoria, Australia. The US study also reported that TNR had the lowest costs (US $90 per cat) because of minimal costs for housing and associated care [65].

Although TNR is illegal in most parts of Australia, stray cats can be legally sterilized, provided the carer becomes the legal owner. In Banyule’s program, none of the semi-owners that enrolled a cat was unable or unwilling to take ownership, which reflected their trust in the AMOs. The program operated without imposing obstacles on the community, and individuals did not fear reprisal of enforcement action. After the initial free cat registration period for the first year, there was no subsequent follow-up for renewal payments. Owners with multiple cats also faced no further action to obtain permits for multiple cats. Nevertheless, when welfare concerns for the cats were evident in multi-cat situations, discussions aimed at reducing cat numbers occurred with owners and semi-owners. The efficacy of these discussions stemmed from the confidence that cat owners and caretakers placed in the AMOs to prioritize the welfare of the animals, thereby enabling them to make informed decisions on what was best for them and the cats in their care.

Given financial advantages for councils to implement the management of free-roaming cats by sterilization, it is recommended that legislative changes be implemented to allow TNR and RTF in Australia, given the benefits on mental health of staff. TNR is particularly valuable where multiple cats are being cared for at a site. Provision in the legislation should be made for the cats to be microchipped as community cats, with contact details provided for a welfare agency, rescue group, or the caregiver. This model is being successfully trialed under a research permit provided by the Queensland Department of Agriculture and Fisheries to the University of Queensland and operated by the Australian Pet Welfare Foundation, in collaboration with RSPCA Qld and Animal Welfare League Qld. Cat intake and euthanasia in the trial suburbs has rapidly decreased over one-to-three years following implementation, depending on the numbers sterilized per 1000 residents and the degree of microtargeting [66].

## 5. Conclusions

In conclusion, the traditional methods of trapping wandering and nuisance cats have not resulted in long-term reductions in cat-related calls to councils. However, following the implementation of a microtargeted free sterilization program for owned and semi-owned cats, marked reductions in cat-related calls, impoundments, euthanasia, and costs were realized, similar to that reported in US programs. It is recommended that urban cat management policies and programs are revised and, instead of being focused on a traditional compliance-based approach, are focused on being assistive, helping owners and semi-owners have their cats sterilized and identified with a microchip. Legislative changes need to be implemented to facilitate this approach to assist people caring for multiple stray cats, instead of the current approach to trap and euthanize most of these cats which are poorly socialized, which is documented to damage the mental health of shelter and pound staff and cat caregivers [4,5,7,50,67]. An assistive method is aligned with a One Welfare approach which balances and optimizes the well-being of animals, people, and their environments [30].

## Figures and Tables

**Figure 1 animals-14-01615-f001:**
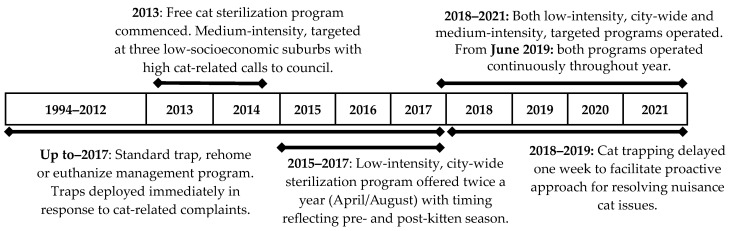
The timeline of the free cat sterilization programs in the city of Banyule from 2013 to 2021. Prior to 2018, residents calling about a nuisance cat were immediately provided a trap cage, and the cat was impounded. If not reclaimed by its owner, it was adopted or euthanized. In 2013/14, a free cat sterilization program was implemented in three target suburbs that were lower socioeconomically (population 13,445) and had the highest rate of cat-related calls and cat impoundments. On average, over the eight years, 4.1 cats/1000 residents were sterilized per year in the three target suburbs and 0.8 cats/1000 residents city-wide.

**Figure 2 animals-14-01615-f002:**
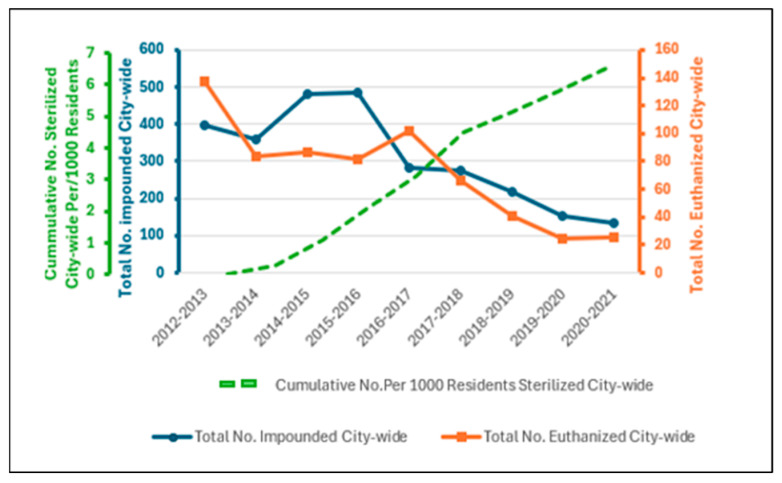
Cumulative numbers of cats sterilized city-wide, and number of cats impounded and euthanized per year following implementation of a free sterilization, microchipping, and registration program for owned and semi-owned cats in 2013/14.

**Figure 3 animals-14-01615-f003:**
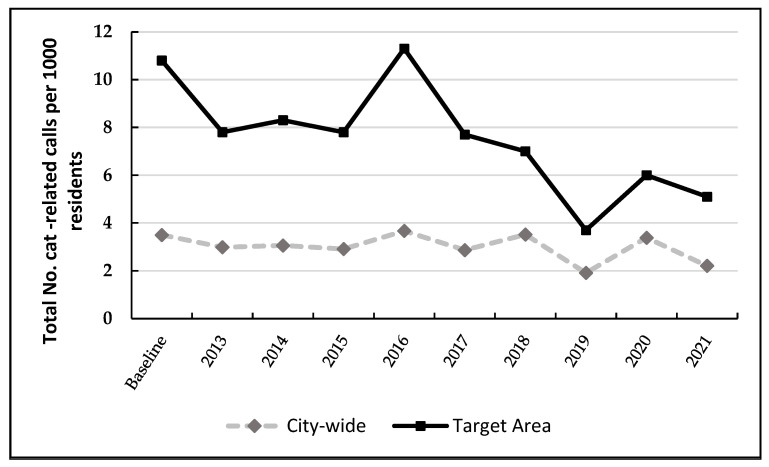
Cat-related calls for found cats and nuisance cats per 1000 residents in 3 target suburbs (lowest socioeconomic indices for city of Banyule) and city-wide from 2011 to 2021 following the implementation of a free sterilization program for cats in 2013 (baseline years 2011 to 2012 averaged 10.8/1000 residents; Table 3).

**Table 1 animals-14-01615-t001:** For financial years 2017/18 to 2020/21, the number of cats (and number per 1000 residents) admitted to Cat Protection Society (CPS) city-wide, including target suburbs, by AMOs (impounded cats); stray cats found by the public; and by owners relinquishing cats.

Year	Cats Impounded City-Wide by AMOs (per 1000 Residents)	Stray Cats from the Public Handed Directly into CPS (per 1000 Residents)	Cats Relinquished by Owners Directly to CPS (per 1000 Residents)	Total Cat Intake into CPS from the City of Banyule from All Sources, Including Target Area (per 1000 Residents)
2017/18	274 (2.1)	315 (2.4)	204 (1.6)	793 (6.1)
2018/19	217 (1.7)	211 (1.6)	102 (0.8)	530 (4.1)
2019/20	152 (1.2)	244 (1.9)	86 (0.7)	482 (3.7)
2020/21	134 (1.1)	228 (1.8)	102 (0.8)	464 (3.6)

Data prior to financial year 2017/18 were not available for strays and owner-relinquished cats and kittens taken directly to CPS because the paper records were destroyed by the facility at the time of transfer to an electronic recording system in 2017. From 2017/18 to 2020/21, there were decreases in city-wide impoundments by 51%, stray cats from the public to CPS by 28%, and owner-relinquished cats to CPS by 50% (Table A3). Australian financial year is 1 July to 30 June.

**Table 2 animals-14-01615-t002:** Number and percentage of cats impounded city-wide (including three target suburbs) by AMOs and their outcomes (reclaimed, rehomed, and euthanized cats) at baseline (2012/13) and for eight years following implementation of a free sterilization program in October 2013.

Year	Number of Cats/Kittens Impounded by AMOs *	Number Reclaimed (% Reclaimed)	Number Rehomed * (% Rehomed)	Number Euthanized * (% Euthanized)	Number Euthanized per 1000 Residents	% Decrease in the Number Euthanized
2012/13	396	23 (6%)	235 (59%)	138 (35%)	1.1	Baseline
2013/14	359	24 (7%)	251 (70%)	84 (23%)	0.7	39%
2014/15	481	37 (8%)	357 (74%)	87 (18%)	0.7	37%
2015/16	487	65 (13%)	340 (70%)	82 (17%)	0.6	41%
2016/17	284	51 (18%)	131 (46%)	102 (36%)	0.8	26%
2017/18 ^^^	274	38 (14%)	170 (62%)	66 (24%)	0.5	52%
2018/19 ^^^	217	24 (11%)	152 (70%)	41 (19%)	0.3	70%
2019/20 ^^^	152	30 (20%)	98 (64%)	24 (16%)	0.2	83%
2020/21 ^^^	134	21 (16%)	88 (66%)	25 (19%)	0.2	82%

* There was an overall reduction (Table A3) in the number of cats impounded, rehomed, and euthanized between 2012/13 and 2020/21. ^^^ Complaint handling changed from immediately trapping nuisance cat/s to an assistive response.

**Table 3 animals-14-01615-t003:** Cat-related calls/1000 residents and costs to council from 2011 to 2021 based on estimated cost to Animal Management Services of AU $290 per call calculated from wage rate of AU $32.80/h for customer service officer (1 h) to process the call, AU $35.90/h for local laws administration to allocate job to an AMO (1 h 5 min), and AU $35.89/h for AMO time (3 h 40 min) plus vehicle costs of AU $86 (Table A2). Sterilization program commenced October 2013, and, hence, 2013 was not used for calculating savings.

Year	Human Population	Cat-Related Calls City-Wide (Including Target Area) (Baseline Average 436 Calls *)				Cat-Related Calls in 3 Target Suburbs (Baseline Average 145 Calls *)			Costs and Savings for Cat-Related Calls for Whole City-Average Baseline Cost @ AU $126,322	
		Total Calls	Cat Related Calls per 1000 Residents	Change in City-Wide Calls	City-Wide % Change	Total Calls	No./1000 Residents	Target Suburbs % Change	Costs AU $ to Council from 2011 to 2021 Based on Estimated Cost	Annual Saving from Baseline in AU $
2011	122,983	462	3.8	NA	Baseline	170	10.8	Baseline	$133,980	Baseline
2012	123,584	410	3.3	NA	Baseline	120	8.9	Baseline	$118,900	Baseline
2013	124,314	371	3	NA	NA	105	7.8	NA	$107,490	NA
2014	125,107	383	3.1	53	12%	111	8.3	23%	$111,070	$15,370
2015	126,088	367	2.9	69	16%	105	7.8	28%	$106,430	$20,010
2016	127,447	468	3.7	−32	−7%	152	11.3	−5%	$135,720	− $9280
2017	128,660	369	2.9	67	15%	103	7.7	29%	$107,010	$19,430
2018	129,645	456	3.5	−20	−5%	94	7	35%	$132,240	− $5800
2019	130,607	250	1.9	186	43%	49	3.7	66%	$72,500	$53,940
2020	130,294	441 ^#^	3.4 ^#^	−5 ^#^	−1% ^#^	81 ^#^	6	44%	$127,890	− $1450
2021	127,370	281	2.2	155	36%	72	5.1	51%	$81,490	$44,950
**TOTAL**		**4258**		**506**		**1162**			**$1,234,720**	**$137,170**

* Baseline is the average of calls in 2011 and 2012 years. NA = not applicable. ^#^ From mid-March to the end of October 2020, residents were in lockdown and confined to their homes because of COVID-associated restrictions.

**Table 4 animals-14-01615-t004:** Cost analysis for estimated savings for financial years from 2013/14 to 2020/21 associated with the implementation of a free cat sterilization program in Banyule city in 2013/14 onward. Costs (in AU $) relate to charges by the service provider (CPS) per cat impounded from the city and costs for AMO and customer service staff time associated with responding to cat-related calls to council.

Year	Impounded Cats Admitted to CPS			Cats Admitted to CPS from All Sources in Banyule				Sterilization Surgeries	
	No. of Cats	Decrease in Cats to CPS	* Costs to Council	No. of Cats ^	Decrease in Cats to CPS	* Costs to Council in AU $	Decrease in Costs to Council AU $	No. of Surgeries	Cost in AU $ #
**2012/13** **(Baseline)**	**396**	**Baseline**	**$31,680**	**1164 ^**	**Baseline**	**$93,120**	**Baseline**	**0**	**0**
2013/14	359	**37**	$28,720	1055 ^	109	$84,400	$8720	40	$6300
2014/15	481	−85	$38,480	1414 ^	−250	$113,120	−$20,000	93	$11,865
2015/16	487	−91	$38,960	1432 ^	−268	$114,560	−$21,440	138	$14,490
2016/17	284	112	$22,720	835 ^	329	$66,800	$26,320	130	$13,650
2017/18	274	122	$21,920	793	371	$29,680	$29,680	177	$18,585
2018/19	217	179	$20,620	530	634	$72,910	$72,910	90	$6300
2019/20	152	244	$22,800	482	682	$72,300	$102,300	90	$3150
2020/21	134	262	$39,300	464	700	$69,600	$105,000	73	$3150
**Totals 2013/14 to 2020/21**	**2388**	**780**	**$233,520**	**7005**	**2307**	**$623,370**	**$303,490**	**831**	**$77,490**

* Estimated costs per cat charged by CPS to the city of Banyule were calculated @ AU $80/cat from 2012/13–2017/18 and AU $150/cat from 2018/2019. Baseline cost in 2012/13 for impounded cats was AU $31,680 (396 cats), and for all sources of cats was AU $92,120 (1164 cats) @ AU $80/cat). ^ Estimated number of cats admitted to CPS from Banyule from all sources (impounded by council AMOs, general public handing in stray or found cats, and owner-relinquished cats). This estimate was based on 2.94 × number of cats impounded by council because impounded cats represented approximately 34% of total cats admitted to CPS in the last 4 years of the program. This calculation was utilized for financial years 2012/13 to 2016/17. # The average cost of sterilizations to the city of Banyule was AU $105 (80 cats @ AU $157.50 charged by RSPCA, 618 cats @ AU $105 charged by local veterinarians). There were 133 free sterilizations included in the total number of sterilizations.

## Data Availability

Relevant data are reproduced in the text.

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
