# Peer review of "Impact of a Local Government Funded Free Cat Sterilization Program for Owned and Semi-Owned Cats"

_animals, 2024, doi:10.3390/ani14111615_

Round 1

Reviewer 1 Report

Comments and Suggestions for Authors

This is a well-written paper that contains a tremendous amount of data on an important topic. I have a few minor suggestions/questions for the authors to consider prior to publication:

You discuss "council pounds" and contrast them with "animal welfare shelters", yet I'm still unclear on the difference here. (Line 49)

In general, the Introduction might benefit from some sections, just as you've done with the Results and Discussion.

Is the program truly named "Trap-Adopt-Kill" as stated a couple of times in the manuscript? I am familiar with TNR programs, but not one quite so literally named.

Related to the above comment, I assume that impounded cats were euthanized or available for adoption. Indicating that cats were "rehomed" makes it sound like all cats that were not euthanized were easily placed in new homes. Is that accurate?

Does "entire" cat mean "intact" cat? One that is not spayed or neutered? (Line 70)

Word tense switches to past tense for some reason in Line 125.

Some methods, such as cost, seem better suited for the results section.

Similarly, some of what was included in Complain Handling, Section 2.3, seems better addressed in the Introduction, as it isn't part of the current methodology.

Sterilization surgeries are discussed twice, in both section 2.4 and 2.5. Is that necessary?

You've provided a number of helpful figures and tables, which is appreciated. One that might be of use to the reader is a timeline for the project, including what was done previously, and what was new and when it was implemented.

Reviewer 2 Report

Comments and Suggestions for Authors

This is a well written manuscript on a very important topic. The authors did a great job outlining the program, giving a clear background on how and why it was implemented, and discussing the merits of the program from the financial and animal welfare perspective.

I have a few minor corrections/suggestions to improve the manuscript listed below. 

Introduction:

Lines 60-61- "In Australia, 3-9% of adults feed daily one or more cats they do not perceive they own, and they feed an average of 1.5 cats" The sentence may read better if it was broken into two sentences. It was a bit hard to read, maybe because there is a lot of information to process in that small sentence. 

Results:

Figure 1- Please clarify the number for "Cumulative No. Sterilized City-Wide" is per/1000 residents. The other two numbers are not based on that and so the figure reads that only 6 cats were sterilized city-wide. 

Line 462- I think you are missing a % sign after the number "39" in this sentence. 

Author Response

"Please see attachment"

Reviewer 3 Report

Comments and Suggestions for Authors

The work raises the critical issue of wandering cats and the problems that result from it. Many countries do not have effective programs for managing stray or free-roaming cat populations. In many countries, there is no obligation to mark (using a microchip) cats and register them. Animals moving without control can be a source of many diseases; they can hunt birds even though they have access to food and get into fights with other owners' cats. In addition, unlimited reproduction is possible if they are not neutered. Therefore, free-living cats are a source of concern for residents.

The solutions presented by the authors can be an inspiration for solving this type of problem in many countries. However, I have some comments.

What does the word sterilization mean, according to the authors? Is it tubal ligation in female cats and vasoligation in male cats which disrupt the hormonal functions of the reproductive organs (ovaries and testicles).Or rather the complete removal of the reproductive organs, i.e. the uterus and ovaries (ovariohisterctomy) in female cats and the testicles in males - in that case it is a castration procedure. The word sterilization is used 123 times, so it would be necessary to explain the cats' treatment in more detail,  especially in the subsection 2.5. Sterilization surgeries, line 293.

The Matherial and Methods chapter is very extensive; it contains data that is later included in the Results chapter - you can skip it because this multitude of information blurs the entire idea of this chapter Conclusion is essentially a repetition of information and data from chapters: the ‘Materials and methods’ and ‘Results’. The actual Conclusion starts from line 851, and the text from line 835 to line 850 can be deleted.

Despite the comments, the work meets the requirements for publication in the journal Animals and can be published after major revision.

Author Response

"Please see attachment"

Reviewer 4 Report

Comments and Suggestions for Authors

A very clear and well researched paper showing impact of a different way of working on prevention of cat problems.  However  Australia has specific legislative restrictions on cat roaming which may not apply in other countries - it would be worth stating what the legislative framework is in the introduction line 51 - in most other countries cats are not impounded by local authorities for a set time period

the final year of the survey 2020-21 was impacted heavily by Covid particularly in Victoria which saw very strict measures on personal movement - there is no mention of Covid and its impact on the 2020-21 data eg Table 1 or under discussion - there needs to be some mention of this and discussion on its impact on the results, if any

Table A2 clearly sets out the costs for one cat related call to the council - however there doesnt seem to be a similar table setting out the costs for the neutering programme and this would be helpful to give a comparison on savings as I assume the CSO tasks and costs would be similar in neutering to those in responding to cat calls even if the neutering itself was supplied free or at low cost by RSPCA 

Author Response

"Please see attached"

Round 2

Reviewer 3 Report

Comments and Suggestions for Authors

thank you for making the corrections